# Assessment of Rheological Models Applied to Blood Flow in Human Thoracic Aorta

**DOI:** 10.3390/bioengineering10111240

**Published:** 2023-10-24

**Authors:** Alexander Fuchs, Niclas Berg, Laszlo Fuchs, Lisa Prahl Wittberg

**Affiliations:** 1Department of Radiology in Linköping, Linköping University, 581 83 Linköping, Sweden; 2Department of Health, Medicine and Caring Sciences, Linköping University, 581 83 Linköping, Sweden; 3FLOW, Department of Engineering Mechanics, Royal Institute of Technology (KTH), 100 44 Stockholm, Swedenlf@mech.kth.se (L.F.)

**Keywords:** thoracic aorta flow, rheological models for blood, effects of blood viscosity

## Abstract

Purpose: The purpose of this study is to assess the importance of non-Newtonian rheological models on blood flow in the human thoracic aorta. Methods: The pulsatile flow in the aorta is simulated using the models of Casson, Quemada and Walburn–Schneck in addition to a case of fixed (Newtonian) viscosity. The impact of the four rheological models (using constant hematocrit) was assessed with respect to (i) magnitude and deviation of the viscosity relative to a reference value (the Newtonian case); (ii) wall shear stress (WSS) and its time derivative; (iii) common WSS-related indicators, OSI, TAWSS and RRT; (iv) relative volume and surface-based retrograde flow; and (v) the impact of rheological models on the transport of small particles in the thoracic aorta. Results: The time-dependent flow in the thoracic aorta implies relatively large variations in the instantaneous WSS, due to variations in the instantaneous viscosity by as much as an order of magnitude. The largest effect was observed for low shear rates (tens s^−1^). The different viscosity models had a small impact in terms of time- and spaced-averaged quantities. The significance of the rheological models was clearly demonstrated in the instantaneous WSS, for the space-averaged WSS (about 10%) and the corresponding temporal derivative of WSS (up to 20%). The longer-term accumulated effect of the rheological model was observed for the transport of spherical particles of 2 mm and 2 mm in diameter (density of 1200 kg/m^3^). Large particles’ total residence time in the brachiocephalic artery was 60% longer compared to the smaller particles. For the left common carotid artery, the opposite was observed: the smaller particles resided considerably longer than their larger counterparts. Conclusions: The dependence on the non-Newtonian properties of blood is mostly important at low shear regions (near walls, stagnation regions). Time- and space-averaging parameters of interest reduce the impact of the rheological model and may thereby lead to under-estimation of viscous effects. The rheological model affects the local WSS and its temporal derivative. In addition, the transport of small particles includes the accumulated effect of the blood rheological model as the several forces (e.g., drag, added mass and lift) acting on the particles are viscosity dependent. Mass transport is an essential factor for the development of pathologies in the arterial wall, implying that rheological models are important for assessing such risks.

## 1. Introduction

Blood is a non-Newtonian fluid mainly composed of water but also containing a wide range of cells, micelles, and molecules of widely different sizes. The multiple functionalities of blood and its components include transporting substances needed by the different organs, being a major player in maintaining an optimal environment (e.g., pH, temperature) in the body as well as defending the body against microorganisms and stopping bleeding. Hence, blood composition may change in terms of numbers and types of cells, micelles and molecules as a response to needs. This adaptive behavior can be rather quick, implying that for all of us, blood rheological properties may vary over the day. In addition, pathological variations of the blood constituents affect blood viscosity. For example, red blood cell (RBC) size distribution (or RBC distribution width (RDW)) has been found to be a predictor of morbidity and mortality (cf Lippi et al. [1], Danese et al. [2], Ananthaseshan et al. [3]). Non-uniform distribution of RBCs within the circulatory system contributes to the difficulty in determining whole blood viscosity (WBV). Blood rheology has been the subject of many studies found in the literature. Often, WBV measurements have been carried out using standard (shear) rheometers (cf Cowan et al. [4]), implying that the flow is laminar and that the shear rate is well defined and solely dependent on the rotation rate (cf Agarwal et al. [5]). Yamamoto et al. [6] used a compact-sized falling needle rheometer on fresh blood samples to measure the relationship between the shear stress (τ) and shear rate (γ). The study identified three typical “regions”: the “Casson” region for a low shear rate range (below 140 s^−1^), the transition region (up to about 160 s^−1^) and the Newtonian fluid region for higher shear rates (above 160 s^−1^). The range of human blood viscosity was found to be in the range of 5.5 to 6.4 mPa s, and 4.5 to 5.3 mPa s for males and females, respectively. Moreover, Wang et al. [7] measured the WBV for a group of healthy individuals showing that both inter- and intra-individual variations were higher in the morning than later in the day for all shear rates. In addition to blood cell content, the types and number of lipoproteins also influence WBV [8].

Over past decades, different types of whole blood viscosity models based on rheological data have been proposed, where power-law-based models were suggested due to the resemblance between blood and other shear-thinning fluids. A recent review of most existing models of whole blood viscosity was presented by Hund et al. [9]. All models include several parameters determined by fitting the models to measurements. For example, Marcinkowska-Gapinska et al. [10] measured the viscosity of 100 whole blood and plasma samples over a range of shear rates (between 0.01 and 100 s^−1^). Applying the measured data to the models of Casson, Ree–Eyring and Quemada, the Quemada model was found to provide the best fit. Gallagher et al. [11] considered models of Bird, Carreau [12], Cross and Yasuda. The aim of the study was to address the problem of inferring model parameters by fitting them to experiments. By refitting published data, families of parameter sets capturing the data equally well were identified. The fitted parameters yielded almost indistinguishable fits to experimental data, but the different parameter sets predicted very different flow profiles. This finding shows the difficulties in representing the fluid physical properties well. To assess the sensitivity of the models, random perturbations were added to the measured data from which new model parameters were derived, showing that the problem was inherent to the models considered. The effects of the Casson and Carreau–Yasuda models on a steady and oscillatory 2D flow in a straight and curved pipe geometry were studied by Boyd et al. [13] using the Lattice Boltzmann method. Differences in velocity and shear were found at low Reynolds and Womersley numbers, although were rather small in terms of velocity profiles. More recently, attempts have been made to improve the low shear stress behavior of rheological models. Jedrzejczak et al. [14] proposed a population balance-based model of blood including hemolysis. The proposed model was compared to the characteristic viscosity of the Carreau Yasuda model for viscosimetric conditions. The model predicted a smoother transition between high and low viscosity zones in a constricted pipe. The model was verified through results from ex vivo experiments. Arzani [15] proposed a hybrid Newtonian and non-Newtonian rheology model by switching to a traditional Carreau–Yasuda model when the residence time exceeded a threshold value. Lagrangian particle tracking was used to detect stagnant regions with increased rouleaux formation likelihood.

Blood flow simulations of clinical interest require considering blood flow in patient-relevant settings. Several studies have assessed and compared different rheological models, providing results displaying discrepancies among the conclusions. Akherat et al. [16] simulated blood flow in arteriovenous (AV) fistula using reconstructed 3D geometries and the models of Quemada and Casson. The results displayed no major differences in the flow field and the flow characteristics. Instead, the shape of the geometry was found to be far more important for the WSS distribution as compared to the effect due to the rheological model. Investigating the flow in a patient-specific aorta model, Karimi et al. [17] applied nine rheological models (three Casson model variants, Carreau, Carreau–Yasuda, Cross, Power-law, Modified Power-law, and Generalized Power-law parameters), focusing on WSS and the deviation of the computed viscosity as compared to a reference viscosity (3.45 × 10^−3^ Pa s). The largest differences in WSS were located near the branches and found more pronounced at a low flow rate (diastole) where the viscosity was considerably larger than the value used in the Newtonian computations. Karimi et al. [17] concluded that various rheological models may yield equally good results apart from the Cross model. A similar conclusion was presented by Johnston et al. [18], who numerically investigated the effects of six rheological models (including a Newtonian) on WSS during the cardiac cycle. The results were comparable for all models under steady flow conditions and at unsteady mid-range flow velocities (around 0.2 m/s). It was reported that a Newtonian model was a good approximation in regions of mid-range to high shear, whereas for low shear rates, the Power-law model was found more appropriate. However, in the numerical study of a pipe flow by Jahangiri et al. [19], the rheological models of Carreau, Carreau–Yasuda, modified Casson, Power-law, Generalized Power-Law and Walburn–Schneck, except the Generalized Power-law model, resulted in graphically the same axial velocity profile and WSS behavior. In contrast, Apostolidis et al. [20], considering the flow in a coronary artery setup, found as much as 50% change in WSS (instantaneous and local) when considering the effect of the Casson model. Mendieta et al. [21] studied the importance of blood rheology in patient-specific simulations of stenotic carotid arteries. Four rheological models (Newtonian and four non-Newtonian models (Carreau, Cross, Quemada and Power-law) were considered. Averaged quantities related to WSS descriptors (such as those used here, Section 3 of the results) were compared. The main conclusion was that the assumption of a Newtonian model is reasonable in terms of the overall flow pattern or the mean values, but a non-Newtonian model is necessary when the low TAWSS region and strongly stagnant flow regions are in focus. Further patient-specific-related geometries were studied by Liepsch et al. [22] to assess the impact of non-Newtonian viscosity models on the hemodynamics of a cerebral aneurysm. The flow in the aneurysm was computed using the Newtonian, Power-law, Bird–Carreau, Casson and Local viscosity models. Although similar flow patterns were observed both for Newtonian and non-Newtonian models, a quantitative comparison performed on a group of monitoring points revealed an average difference between the Newtonian and non-Newtonian models of about 12%. Furthermore, in low-speed regions, the differences were even larger (20% to 63%). Skiadopoulos et al. [23] considered Newtonian, Quemada and Casson blood viscosity models for simulating pulsatile flow in patient-specific geometry of the iliac bifurcation. The effect of the rheological models was monitored through the WSS distribution, magnitude and oscillations and viscosity behavior as a function of the shear rate. In addition to the commonly used WSS-related indicators (OSI and TAWSS), the study considered the (wall) area averaged WSS and the corresponding area averaged shear rate. The magnitude of the WSS and its oscillations were found to depend on the shear rate and the rheological model. The main conclusion of Skiadopoulos et al. [23] was that the Newtonian approximation is mostly applicable for high shear and flow rates. The Newtonian model was found to overestimate the possibility of the formation of atherosclerotic lesions in regions with oscillatory WSS. In a recent paper, Mendieta et al. [21] compared the effect of rheological models on the WSS-related parameters in stenotic carotid flows. Four rheological models were considered (Carreau, Cross, Quemada and Power-law). The largest differences between the Newtonian and non-Newtonian models were noted for an OSI of about 12% (in terms of maximum and mean values). Regarding the TAWSS, the difference was less than 6%, except for the Quemada model where the difference was as much as 26%. The authors concluded that the assumption of a Newtonian model can be reasonable; however, non-Newtonian models were found necessary in low TAWSS regions. Mirza and Ramaswamy [8] observed that the WSS using Newtonian and non-Newtonian models differ noticeably.

Most of the above cases considered laminar and/or transitional flow regimes. In contrast, Molla and Paul [24] studied a turbulent flow in a channel with a constriction, computed by Large Eddy Simulations. Five rheological models (Power-law, Carreau, Quemada, Cross and a modified Casson) were compared in terms of peak shear rate, mean shear stress and pressure, re-circulation zones and turbulent kinetic energy. The main finding was that the non-Newtonian viscosity models extended the post-stenotic re-circulation region and reduced the turbulent kinetic energy downstream of the stenosis. In the simulations, the shear rate was limited to lower than 100 s^−1^, i.e., in the range where the viscosity differs strongly from the high shear rate range which in turn implies higher levels of viscosity and a lower level of turbulent kinetic energy (TKE). In terms of TKE, the differences between the rheological models were rather modest.

Rheological blood models are calibrated to in vitro measured data. However, when applied to patient-specific simulations, the main goal of the models is to be able to account for important fluid physics and capture relevant clinical observations. Despite the consensus that the importance of non-Newtonian rheological effects is more pronounced for low shear rates, the conclusions with respect to the importance of modeling the non-Newtonian effect may differ depending on the application. Hence, the aim of the current study was to provide some general principles for when to employ Newtonian or non-Newtonian rheological models for blood flow simulation of clinical interest. We show how several of the commonly used parameters to quantify blood flow characteristics depend on the chosen rheological model and the origin of these dependencies.

## 2. Materials and Methods

The geometry of the thoracic aorta was derived from a computed tomography angiography (CTA) study of a healthy patient (Figure 1a). The computational domain consisted of the ascending aorta, the aortic arch, the descending aorta and the three main branching arteries: the brachiocephalic (BC) artery, the left common carotid artery (LC) and the left subclavian artery (LS)). The blood was assumed to be incompressible with constant bulk density and fixed concentration of red blood cells (RBC). The effects of other blood components were neglected. Thus, the blood was modeled as a non-homogenous mixture satisfying conservation of mass (Equation (1a)) and momentum (Equation (1b)).
(1a)$$\frac{\partial \rho u_{i}}{\partial t}+\frac{\partial (\rho u_{i}u_{j})}{\partial x_{j}}=-\frac{\partial p}{\partial x_{i}}+\frac{\partial }{\partial x_{j}}\mu \frac{\partial u_{i}}{\partial x_{j}}$$
(1b)$$\frac{\partial \rho }{\partial t}+\frac{\partial \rho u_{i}}{\partial x_{i}}=0$$
where and *µ* are the density and viscosity of the mixture, respectively, *p* is the pressure and *u_i_* is the Cartesian velocity component in the *i*:th direction. The bulk viscosity depends on the local shear rate, accounted for by applying the following rheological models: Newtonian fluid (viscosity, *µ* = 3.35 mPa s and density *ρ* = 1102 kg/m^3^) and three non-Newtonian models of Walburn and Schneck [25], Casson [26] and Quemada [27,28] to account for the local mixture viscosity, *µ = µ_eff_*. The hematocrit value was set to a fixed value of α = 0.45.

### 2.1. Mixture Viscosity Models

The simplest non-Newtonian model, used in multiple publications and available in several common pieces of software, is the Power-law model of Ostwald de Waele [6]. In its simplest form, the model directly relates effective viscosity (m_eff_) to the shear rate $\mu_{eff}=k\gamma^{n-1}$, where *γ* is the magnitude of the shear rate tensor $\gamma_{ij}=\frac{1}{2}\left(\frac{\partial u_{i}}{\partial x_{j}}+\frac{\partial u_{j}}{\partial x_{i}}\right)$. The power of the shear rate is negative for shear-thinning fluids (*n* < 1). The three considered non-Newtonian models were as follows:The Walburn–Schneck [25] model for the effective viscosity (*µ_eff_*) of the blood (mixture), which is an extension of the Power-law model. In the current work, using *α* = 0.45, the effective viscosity is modeled by:
(2)$$\mu_{eff}=\max(0.0034,\gamma^{-0.0225})$$

The Casson [26] model, based on a calibrated Power-law concept. The following form and parameters were used:


(3)
$$\mu_{eff}=\min\left(\mu_{max},\frac{{\left[\sqrt{k_{C}(\alpha )\gamma }+\sqrt{\tau_{y}(\alpha )}\right]}^{2}}{\gamma }\right)$$


The model parameters were derived for a hematocrit of *α* = 0.45 (Cokelet et al. [29]; Perktold et al. [30]).
The Quemada model [27,28]. The model may include two variables: the hematocrit (*α*) and the shear rate (*γ*). Here, a constant *α* = 0.45 was used leading to a simpler formulation:
(4)$$\mu_{eff}=\mu_{p}{\left(1-0.225k(\gamma )\right)}^{-2}$$
with plasma viscosity *µ_p_* = 1.32 mPa s. *k* is an expression of *γ* (further details are found in Fuchs et al. [31].

### 2.2. Boundary Conditions

Regarding boundary conditions, no-slip conditions were set on the (rigid) walls of the thoracic aorta (Figure 1a). The outlet sections were extended to avoid the effects of the outlet conditions on the domain of interest. The exit planes were the brachiocephalic artery (BC), left common carotid artery (LC), left subclavian artery (LC) and the exit plane of the thoracic aorta (EXT). At the inlet, a time-dependent flow rate profile derived from a measured human cardiac profile was imposed (Figure 1b), representing 90 heartbeats per minute (BPM) and 9 L per minute (LPM). This inflow condition was chosen since the flow profile induces temporal and spatial gradients. The axial inlet velocity vector was uniformly distributed across the inlet plane. The other components of the velocity vector were set to 0. The main branches were extended so that no flow recirculation occurred at any time during the cardiac cycle. The flow rate in the main branches, BC, LC and LS was set to 15%, 7.5% and 7.5%, respectively (Benim et al. [32]). The flow out from the thoracic aorta (EXT plane) was set to 70%. The computational domain was discretized by about 5 million computational cells and found to yield adequately accurate results.

The governing equation was discretized using a formally second-order finite-volume scheme. The discrete equations were advanced in time using an implicit solver (OpenFoam 5.0). The results were processed using MATLAB, Paraview, VTK and our own python scripts.

### 2.3. Viscosity Models and WSS Sensitivity Indicators

#### 2.3.1. Monitoring Effects of Non-Newtonian Viscosity

The different non-Newtonian models lead to blood mixture viscosity coefficients varying in space and time. To quantify the variation of the viscosity, the following indices were used (Johnston et al. [18]): the relative viscosity, *I_L_*, and the non-Newtonian importance factors, *I_g__-space_* and *I_g__-time_* capturing the space and time effects, respectively.
(5)$$I_{L}=\frac{\mu }{\mu_{ref}}$$
(6)$$I_{g-space}(t)=\frac{1}{N}{\frac{\left[\sum_{i=1}^{N}\mu (x_{i},t)-\mu_{ref}{)}^{2}\right]}{\mu_{ref}}}^{1/2};\;I_{g-time}(x)=\frac{1}{M}{\frac{\left[\sum_{j=1}^{M}\mu (x,t_{j})-\mu_{ref}{)}^{2}\right]}{\mu_{ref}}}^{1/2}$$
where m_ref_ is the reference viscosity (*µ_ref_* = 3.5 mPa s, used for the Newtonian case).

#### 2.3.2. WSS-related Indicators

The WSS plays a major role in the processes in the arterial wall and depends directly on the near-wall viscosity. To assess the impact of the rheological models, different WSS indicators were used.
WSS(*x*,*t*) and TD_WSS(*x*,*t*)
(7)$${\mathrm{WSS}}_{i}(x,t)=|n_{j}\tau_{ij}|\;\tau_{ij}=\mu \gamma_{ij}\;\mathrm{TD}\_\mathrm{WSS}(x,t)=|\frac{\partial \mathrm{WSS}}{\partial t}|$$
with *n_j_* being the wall *j*-th component of the wall normal vector.

Time average-based expression of WSS: TAWSS, OSI and RRT.

Time-averaged Wall Shear Stress (TAWSS) cf. Suo et al. [33] and Chen et al. [34], a local time-averaged WSS:(8)$$\mathrm{TAWSS}=\frac{1}{T}\int_{0}^{T}\left|{\mathrm{WSS}}_{i}\right|dt$$

Spatial variation of WSS provides an indication of the WSS level and its spatial non-uniformity (i.e., WSS gradient), but does not contain any information about WSS temporal variation.

Oscillatory Shear Index (OSI) (cf He et al. [35], Chen et al. [34]), a measure for temporal sign change in WSS vector. The OSI is defined as
(9)$$\mathrm{OSI}=\frac{1}{2}\left(1-\frac{\left|\int_{0}^{T}{\mathrm{WSS}}_{i}dt\right|}{\int_{0}^{T}\left|{\mathrm{WSS}}_{i}\right|dt}\right)$$

Hence, the OSI varies between 0 and 0.5. When WSS_*i*_ has a constant sign (possibly oscillatory but maintaining the same sign of the WSS vector), OSI = 0. When WSS_*i*_ changes signs such that the integral of the positive and negative sequences are equal, the OSI gets the value 0.5. Values of 0 < OSI < 0.5 indicate a sign oscillatory WSS.

The Relative Residence Time (RRT) has been formulated (cf Rikhtegar et al. [36], Gallo et al. [37]) as follows:(10)$$\mathrm{RRT}=\frac{1}{(1-2\cdot \mathrm{OSI})\cdot \frac{1}{T}\int_{0}^{T}\frac{\left|{\mathrm{WSS}}_{i}\right|}{\mu }dt}$$

## 3. Results

To elucidate the differences and similarities of the flow and shear stress characteristics due to the different rheological models, the results are presented in terms of the following parameters within the lumen or near the aortic wall:
The relative size of the viscosity coefficient, *I_L_*, and the so-called non-Newtonian importance factor, *I_g_* (space and time).The impact of the rheological model on
○The WSS and its time derivative (TD_WSS)○WSS-related indicators: OSI, TAWSS and RRT. The results are presented in the form of spatial and/or temporal distribution of the different parameters○The extent of retrograde flow particle transport in the thoracic aorta, expressed in terms of residence times.The extent of shear stress magnitude below 100.

### 3.1. Viscosity Coefficient

As the hematocrit is kept constant, the considered rheological models depend only on the shear rate. Figure 2a depicts the mixture viscosity of the four rheological models. Behaving as Newtonian fluids at large shear rates and differing mainly at low shear rates, differences between the rheological models are expected for local shear rates below the order of 100 s^−1^. A direct comparison of the contributions of the different models is displayed in terms of *I_L_* and *I_g_*, Equation (5), and shown in Figure 2a,b. The space averaging of the normalized viscosity (*I_L_*) and the non-Newtonian importance factor enables the assessment of the temporal variations of the rheological models during the cardiac cycle. Obviously, for the Newtonian model *I_L_* = 1 and *I_g_* = 0 and not shown. The Walburn–Schneck model yields the largest values as compared to the other two models. The Casson model presents a mean increase in viscosity only by a factor less than 1.4 whereas the corresponding value for the Quemada model is about 1.8. All three models have the largest viscosity in early systole, decreasing during systole and increasing at late diastole. This behavior reflects the shear-thinning property. In systole and early diastole, the shear rates are largest, leading to lower viscosity whereas, with a lower shear rate, the viscosity increases. The importance factor (*I_g_*) also shows that the deviation from the Newtonian reference value is smallest for the Casson model and largest for Walburn–Schneck. The local minima are found around 0.1 s and 0.45 s, Figure 2c, and correspond to the instantaneous peaks of flow rate and largest shear (Figure 1b). The local peak at about 0.2 s is linked to the strongest flow rate deceleration inducing retrograde flow and temporarily decreasing shear rates in parts of the domain. A similar effect is observed in the late diastolic phase.

An insight into the behavior of the rheological models for the aortic flow under consideration is obtained by considering the probability distribution (PDF) of *I_L_* and *I_g__-space_*. The PDFs for the rheological models of Quemada, Casson and Walburn–Schneck are depicted in Figure 3. The peak probability of the relative time-averaged viscosity is about 1.6, 1.25 and 1.7 for the Quemada, Casson and Walburn–Schneck models, respectively. The corresponding highest probability for *I_g__-space_* is at about 0.5, 0.2 and 0.5, respectively. *I_L_* values close to unity and correspondingly *I_g__-space_* close to 0 imply values close to the reference (Newtonian) viscosity. The peaks shown in Figure 3a indicate that during the cardiac cycle, there are regions where the viscosity is lower than that of the reference value due to the Quemada model. At low shear rates, the models yield considerably larger viscosity values as compared to the reference viscosity. This leads to *I_L_* > 1 and larger *I_g__-space_* values. The largest viscosity values, although associated with low probability, are noted for the Walburn–Schneck and Quemada models. The *I_L_* distributions show that the Casson model has a smaller “signature” relative to the reference value, though the level of fluctuation is relatively large. The Walburn–Schneck distribution indicates that the flow under consideration has a relatively large volume (number of computational cells) with low shear. This effect is less observable in parameters based on time- and space-averaged parameters.

A quantitative but less detailed comparison of the two viscosity parameters *I_L_* and *I_g_*, is given in Table 1. The table shows the mean, standard deviation (std), peak and minimal values of these parameters. The largest peak viscosity value was found with the Walburn–Schneck model. The Casson model shows the smallest mean values but a relatively large peak value.

### 3.2. Model Impact on WSS-Related Quantities

Atherosclerosis is an arterial wall process. Viscous effects are important near the walls and the wall shear stress (WSS) is known to have a significant role in the formation of wall pathologies. The time-averaged probability distributions of the WSS and its temporal derivative (Equation (7)) show similar probability distributions with the different rheological models. Some small differences in terms of the peaks are noted in the WSS (about 2 Pa) and its time derivative (about 20 Pa/s). A common observation is that most of the aortic wall is subjected to a low WSS and only a smaller portion of the wall is subjected to a large and oscillatory WSS. The statistics of the WSS and its time derivative are provided in Table 2. Comparing the models, the mean values of WSS, about 2 Pa, and spatial WSS fluctuation (noted in terms of root mean square (RMS)) are similar. The time derivative of the WSS shows larger differences, where the Walburn–Schneck model displays the largest mean and RMS values. The near-wall behavior of the model is a possible explanation of the larger dissipation generated by this model during systole. The peak of the time derivative of the WSS is smallest for the Casson and the Quemada models, also reflected in Table 2.

The instantaneous WSS in the thoracic aorta is depicted in Figure 1a. The TAWSS (Equation (8)), OSI (Equation (9)) and RRT (Equation (10)) have been associated with the formation of atherosclerosis. Implicitly, these markers (parameters) include blood viscosity and are thus affected by the rheological models. The results show strong similarity in the markers for all four rheological models. The OSI has mostly values below about 0.1 apart from some well-localized regions. Larger OSI values (close to 0.5, indicating sign oscillations of the WSS vector) are seen in the inner wall of the ascending aorta, in the aortic sinus and near the branches of the aortic arch. The RRT indicates analogous behavior with the largest values at the same locations as the OSI. The TAWSS displays stronger values at the junction between the aortic sinus and the ascending aorta and proximal parts of the brachiocephalic artery. In terms of the averaged parameters, the differences between the models are rather small and require close examination of the non-averaged data for making a qualitative assessment.

Visualizations of wall indicators have commonly been adopted in the literature. Here, a different way of assessing the differences between the models is proposed, considering the probability distribution of the OSI, RRT and TAWSS. The OSI probability is largest for a low OSI, where the distribution of probability clearly depends on the rheological model. Lower OSI values indicate the change of sign in the WSS. The Newtonian and the Walburn–Schneck models yield a larger probability for an OSI < 0.1 as compared to the other two models. Thus, the Newtonian and Walburn–Schneck models have larger rates of sign change in the resulting WSS. For a larger OSI (i.e., >0.3), the probability level is similar for the different models, an effect occurring in surface regions without retrograde flow. The TAWSS and the RRT probabilities are insensitive to the viscosity model used, both in terms of values and distribution.

Table 3 summarizes the quantitative statistical data for the three WSS-based markers. The mean and RMS values of the OSI, TAWSS and RRT are close to each other for the different models. It is evident that the specific rheological models are less important for these markers. As the markers are averaged in space and time, the markers may not correctly reflect the impact of the WSS on the arterial processes. Therefore, more sensitive markers must be developed to explain and monitor the formation and progress of arterial wall remodeling.

### 3.3. Impact of Viscosity on Retrograde Flow

A less common measure of the impact of blood rheology on the flow is using the volume of retrograde flow in the thoracic aorta. Two such measures can be defined: (1) based on the direction of the axial velocity compared to the direction of the aortic centerline and (2) based on the (negative) direction of the wall shear stress (i.e., wall parallel component) relative to the centerline direction. The volume and negative WSS of retrograde flow are normalized by the total aortic volume and aortic surface, respectively. The average and maximal values of these quantities are given in Table 4. The time- and space-averaged values show that the amount of retrograde flow is relatively large, about 10% (mean) and about 35% (peak). However, comparing the different models, the values are similar, only differing by about 2% for the averaged and less than 1% in terms of peak relative retrograde volume. The corresponding values for the WSS < 0 are about 7% and 2%, respectively.

Figure 4 depicts the temporal variation of the relative retrograde flow volume and the corresponding relative negative WSS. The peak values of retrograde flow, with both measures, occur at the end of systole (about 0.35 s). The second peak occurs at the flow maximum during diastole (about 0.55 s). The reason for this effect is the same as retrograde flow in cyclic pipe flow (Womersley effect) and is explained in more detail in Fuchs et al. [31,38]. As seen, the differences between the four rheological models are rather limited. One may also note that most differences are found in late diastole.

### 3.4. Shear Rate (SR) and Rheology Model Extent during Cardiac Cycle

The extent of low shear rate (<100 s^−1^) within the thoracic aorta as a function of the cardiac cycle is required to enable the understanding of the consequences of the rheological modeling on, for example, particle transport (discussed in the next section). Figure 5 depicts the probability distributions (counts) for the range of shear rates (SR, 0 s^−1^ to 1000 s^−1^) and the kinematic viscosity (2 × 10^−6^ to 1 × 10^−5^ m^2^/s) for the four rheological models and three time instances in the cardiac cycles: T = 0.12 s (peak systole), 0.33 s (end-systole) and 0.5 s (mid-diastole). Maximal counts in all four cases are observed at low shear rates (below 100 s^−1^). For the three rheological models, it is noted that the width of the distribution is wider for a SR above 200 s^−1^ to 1000 s^−1^. At end-systole, the distribution SRs for the three rheological models show the presence of very low values of SR (about 15 s^−1^) and a second peak at about 65 s^−1^. At mid-diastole, the three rheological models have a narrower distribution showing larger counts at a low SR. In contrast, in the distributions of the Newtonian case, the SR distribution is narrow at systole and wide in mid-diastole. The wider distribution of the SR at T = 0.12 s for the three models leads to a peak in kinematic viscosity near 4 × 10^−6^ m^2^/s due to the presence of a low SR. The second peak and the tail in the kinematic viscosity (at T = 0.12 s) is due to the low SR values (peaks at around 35 s^−1^) for the three rheological models. The main differences between the three models are observed in the distribution widths of the kinematic viscosity: they are widest (3 × 10^−6^ to 8 × 10^−6^ m^2^/s) for the Walburn–Schneck during diastole (T > 0.33 s), whereas the Casson model has a narrower distribution (3 × 10^−6^ to 5 × 10^−6^ m^2^/s) and the Quemada model is in between the two others.

### 3.5. Impact of Viscosity Model on WSS and Its Temporal Derivative

The impact of blood rheology on the WSS and its temporal derivative is determinantal for the development of vascular disease. Thus, the surface average of the WSS and its temporal derivative as a function of the cardiac cycle was studied to shed light on the dynamics of these quantities. Figure 6 depicts the instantaneous and surface mean values of the WSS and its temporal derivatives during the cardiac cycle for the four different rheological models. Note that the geometry in Figure 6 excludes the branching arteries from the aortic arch in contrast to the averages in Table 2 and Table 3. The instantaneous WSS with the four rheological models is shown in Figure 6a (near peak systole). Although the figures are qualitatively similar, a clear local variation in terms of peak values is shown. These variations reduce significantly when taking the average over the whole aorta wall, an effect clearly demonstrated in Table 2 and Table 3. The temporal development of the surface averaged WSS during the cardiac cycle is depicted in frames Figure 6b. The shapes of the curves are similar though the amplitudes differ: the Quemada model has a peak larger than 1.6 Pa compared to about 1.4 for the Casson and Walburn–Schneck models. The instantaneous and surface averages of the temporal derivatives of WSS are depicted in Figure 6c,d respectively. The rheological model has a clear qualitative impact in the ascending aorta and near the branching of the left subclavian artery. The surface-averaged behavior (Figure 6d) shows non-monotone behavior in systole. There is a dip during the initial acceleration in the systole, followed by a quick increase and oscillations as the flow rate is reduced after peak-systole. The initial dip reflects the reduction in viscosity following the initial increase in the shear rate. As the flow increases further, the near-wall shear layer becomes sharper, leading to an increase in the WSS. A more detailed study of the shape of the shear layer shows that after the peak in the systole, multiple inflection points in the axial velocity profiles may be observed (not shown here). Some of these inflection points may lead to instability with exponential growth which is believed to be related to the observed oscillation in the temporal derivative of the WSS. The dynamics of the flow allows a short time for any growth in the oscillations, which are dissipated during diastole. The oscillations in Figure 6d contain different frequencies: higher frequencies for the Newtonian and WS cases and lower frequencies (smoother behavior) for the Quemada case. The observed oscillations in the averages weakly reflect the considerable variation in the spatial distribution of the WSS and its temporal derivatives.

### 3.6. Impact of Viscosity Model on Particle Transport

To assess the accumulated effects of the local viscosity, the impact of blood rheology on particle transport in the thoracic aorta was considered. The residence time of macromolecules, cells and thrombus/emboli plays a central role in certain pathological processes (such as thrombus, formation, growth and transport). In each time step, two sets of spherical particles (2 mm and 2 mm, both with a density of 1200 kg/m^3^) were randomly injected at the inlet (aortic valve) plane. Solid, spherical particles were tracked by integrating Newton’s second law in time, assuming models for the forces acting instantaneously on each particle. Hence, the particle paths reflect the accumulated effects of forces acting on them. It is assumed that the particles are subject to drag, (Saffman’s) lift, added mass and pressure forces. The local fluid viscosity enters in the former three force terms, thereby directly affecting the particle histories. The injected particles were tracked over 10 cardiac cycles. The total number of particles was between 400,000 and 3,000,000. The particle motion and history were characterized by (i) the particle residence time (“age”) in the flow field and (ii) the shortest particle exit times.

Table 5 shows the “age” of particles leaving the four exits from the thoracic aorta after ten cardiac cycles. Particle ages correspond to the time the particle spent in the thoracic aorta since the instance of particle injection. The mean and standard deviation (std) values for both particle sizes are close to each other for each exit and the different rheological models. Similarly, the averages of particle ages (mean values in the table) are similar for each particle group. The differences between the mean values for the two particle groups are also relatively small. The differences in mean values are smaller the longer the particle paths are, measured from the inlet to the exit planes (Table 6, Table 7 and Table 8). However, the rather large std values indicate a significant spread in the data. The shortest particle residence time (denoted by “min” in the tables) is about one cardiac cycle before particles reach the exit plane of the thoracic aorta (EXT). The effect of the rheological models in the shortest residence time is small for each of the exit planes except for the BC exit. The shortest residence time for the 2 mm particles with the Newtonian model is twice as long as for the Quemada and Casson models. This difference is not observed for the larger particles. It may also be noted that for 2 mm particles the mean values are similar and have a relatively large variance. For the LC exit and the larger particle size, the Walburn–Schneck yields a significantly (about 1.5 times) longer minimal residence time as compared to the other rheological models. The same model also deviates (about 60%) from the other models for particles existing in the BC.

Long residence times are of clinical significance as they may affect tendencies for thrombus formation or its transport. The effect of the rheological model on the maximal residence time is larger for the larger particles with the Quemada model and the BC exit. The same model yields the longest residence time for the LC and LS exits. Moving into the thoracic aorta, the particles are subject to varying viscosity, with the particle ages reflecting an averaging effect. This aspect is demonstrated by the results for the exit plane EXT: the particles stay in the domain for almost five and six cardiac cycles for the larger and smaller particles, respectively. The corresponding differences between the models are therefore less than 5% and 20%, respectively.

## 4. Discussion

Studies related to the importance and significance of different rheological models for blood flow simulations have been discussed extensively in the literature. The conclusions do not fully agree in terms of the importance of using non-Newtonian models versus using a constant viscosity value. Several authors have concluded that the flow was significantly affected by certain rheological models and raised a caution (Jahangiri et al. [19]). Other studies reported that the computed flow field general characteristics exhibit no major differences when using Newtonian and non-Newtonian models (cf. Akherat et al. [2]), while other publications find the Newtonian approximation reasonable for high shear flows (Skiadopoulos et al. [23]). The significance and importance of the models have been determined by the visual judgment of graphical data (e.g., surface values of the OSI, TAWSS and RRT). In this study, several different parameters were used to assess the impact of four rheological models to provide clear indications of in which ***situations*** the rheological approach is mostly influential. The forces acting on the endothelium through the WSS (and most importantly its space and time derivatives) are known to affect the development of arterial wall pathologies. Similarly, mass transport (such as platelets, VWF and lipoproteins), which represents the accumulated effect of the local rheology, plays a central role in the development of wall pathologies and in the risk of the formation and transport of thrombi/emboli.

The direct comparison of the relative viscosity and the RMS deviation from a reference (Newtonian case) value shows that the Walburn–Schneck model yields the largest viscosity values throughout the cardiac cycle, particularly during flow acceleration with the lowest values during flow deceleration. The latter effect is directly related to the formation of retrograde flow in a substantial part of the thoracic aorta. The importance factor (*I_g_*) reaches the lowest values at local peaks in flow rate (i.e., peak systole and a local peak at early diastole). The probability distributions of the time-averaged *I_L_* and *I_g_* show clear differences between the non-Newtonian viscosity models. Moreover, the Casson model has the lowest increase in viscosity (near the peak with a smaller range) followed by the Quemada model. The largest increase in viscosity as compared to the reference value is noted for the Walburn–Schneck model. The impact of the models on the temporal development of the space-averaged kinetic energy and the corresponding viscous dissipation rate shows small differences among the models (not shown in the text). This effect may be observed for larger blood vessels but not for vessels where the relative wall region is large. The near wall region is characterized by slow flow and for small vessels the shear rates may be below 100 s^−1^, requiring more refined rheological modeling. The vorticity in the aorta, WSS and its temporal derivative are affected by viscosity and/or alter the blood composition. However, WSS-related quantities (such as the OSI, TAWSS and RRT) are computed as time and averages, which “filters out” any significant differences among the rheological models. Averaged WSS-related parameters may be misleading for assessing the impact of viscosity and in particular time-dependent viscosity on the endothelium and arterial wall re-modeling.

The impact of the rheological models must be evaluated for relevant parameters at the blood flow of relevance, namely the geometrical conditions, boundary conditions, including flow rate variations and the specific aims with the numerical simulations. In the thoracic aorta, the flow is characterized by regions with relatively high shear rates, mostly greater than the order of 10 s^−1^. For large shear rates (depicted in Figure 2a), the differences among the models are considerably smaller than for shear rates below O (100 s^−1^). Hence, for predominantly high shear rate flows, assuming that the blood mixture is a Newtonian fluid is a natural choice. The sensitivity of the rheological models (i.e., the derivative of viscosity with respect to shear rate) is less than 0.01 Pa s/Pa for the Quemada model and shear rates in the range of 100 s^−1^ to 1000 s^−1^. In contrast, the sensitivity of the Walburn–Schneck model is an order of magnitude larger with shear rates of about 100 s^−1^ (not shown here). The increase in viscosity due to the different non-Newtonian models may be as much as an order of magnitude. However, vorticity generation within the aorta is dominated by vortex stretching and with only a marginal, if at all, contribution of viscosity. The WSS, on the other hand, is directly dependent on the local value of the viscosity. Hence, the viscosity model can be expected to be mostly important for medium and smaller size arteries where the flow rate and total wall area are relatively high, hence viscosity plays a more important role compared to large arteries. This effect is clearly shown in arteries often subject to atherosclerosis, such as the coronary, carotid and renal arteries, as shown by van Wyk et al. [39] for a generic aortic bifurcation. For smaller arterioles, the shear rate is low and flow losses are viscosity dominated, hence the impact of the rheological models could be more important than for large arteries. Flow stagnation, independent of the size of the artery, may also require using non-Newtonian rheological models.

Another situation where viscosity plays a central role is in computing the forces acting on particles immersed in the flowing blood. Viscosity enters the formulation of some of the forces acting on small (spherical) particles. These include (viscous) drag, shear-dependent lift (Saffman’s lift force) and added mass. As discussed above, the rheological models may affect the residence times of small particles. Long residence times are relevant for assessing the risk for thrombus formation. Particle transport is also of interest in assessing the risks of embolus transport from the thoracic aorta into vessels leading to the brain. This paper shows the impact of blood rheology on transport and indirectly also on the mixing processes in the thoracic aorta. Prolonged residence time has been believed to have an impact on the metabolism in the endothelium. Similarly, the instantaneous WSS and its temporal variation are believed to affect the endothelium. Unfortunately, the amount of experimental and clinical quantitative data on these processes is very limited and hence it is difficult to assess particular rheological models in these respects.

The results above are limited to a single geometry, single hematocrit and a single heart rate and flow rate. The extent of retrograde flow and particle residence times depends strongly on the geometry of the thoracic aorta and the geometrical details of the arteries branching from the aortic arch and the cardiac output. Therefore, the results and conclusions related to the impact of rheological models as reported here are of qualitative value. However, the conclusions are generally valid whenever viscosity has an accumulative effect (arterial wall remodeling, and transport of blood components). On the other hand, it is well established that time- and space-averaged quantities are not appropriate markers/descriptors for clinical applications due to low specificity. A major limitation of the rheological models used herein is due to neglecting the viscoelastic response of cells and macromolecules. Linderkamp et al. [40] noted that RBC deformability leads to lowering blood viscosity. They also found that RBC (extensional) response time is of the order of 0.1 s, implying a strong interaction between RBC cell response and the temporal variations in the aortic flow, especially during diastole (low shear rate).

## Figures and Tables

**Figure 1 bioengineering-10-01240-f001:**
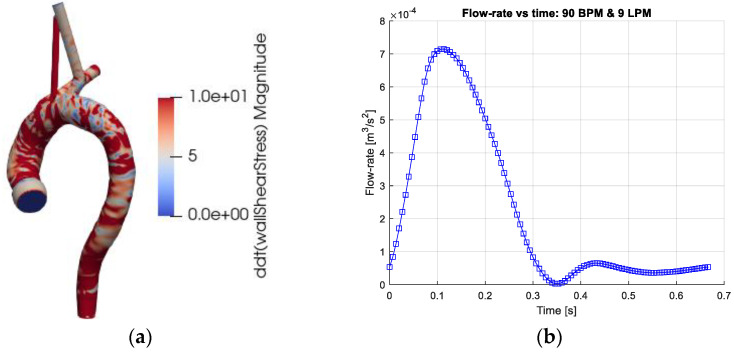
(**a**) The computational geometry of the thoracic aorta. The figures depict the absolute values of time derivative of the wall shear stress (WSS) at the end of systole (time = 0.333 on the right frame). (**b**) The Cardiac output (flow rate) as function of the cardiac cycle. The cardiac output corresponds to 9 L/min (LPM) and heart-rate of 90 beats/min (BPM).

**Figure 2 bioengineering-10-01240-f002:**
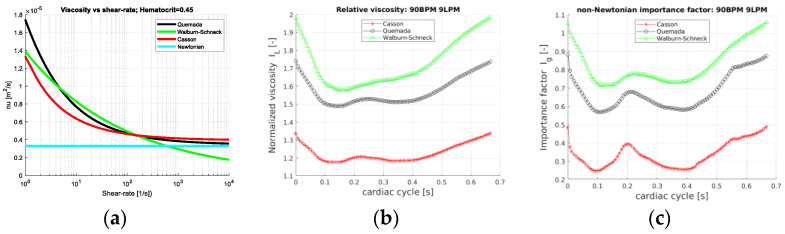
(**a**) Mixture viscosity versus shear rate for the three non-Newtonian models (Equations (2)–(4)), for a = 0.45. (**b**) The space-averaged normalized viscosity *I_L_*(*t*) and (**c**) the importance factor (*I_g-space_*(*t*)), plotted over the cardiac cycle for 90 BPM and 9 LPM.

**Figure 3 bioengineering-10-01240-f003:**
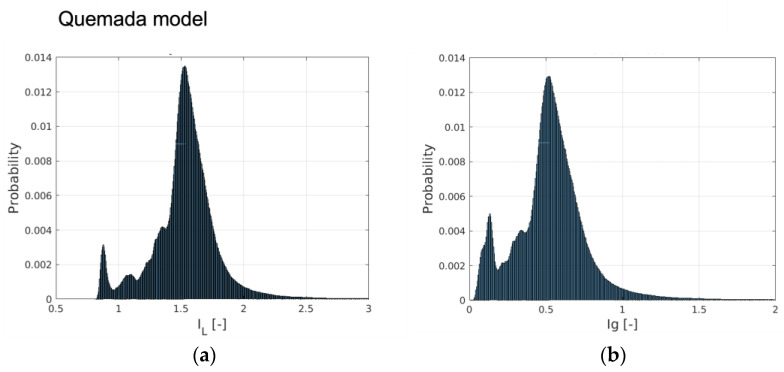
The probability distribution of the relative viscosity (*I_L_*) and importance factor (*I_g-time_*) computed by (**a**,**b**) the Quemada model, (**c**,**d**) Casson and (**e**,**f**) Walburn–Schneck.

**Figure 4 bioengineering-10-01240-f004:**
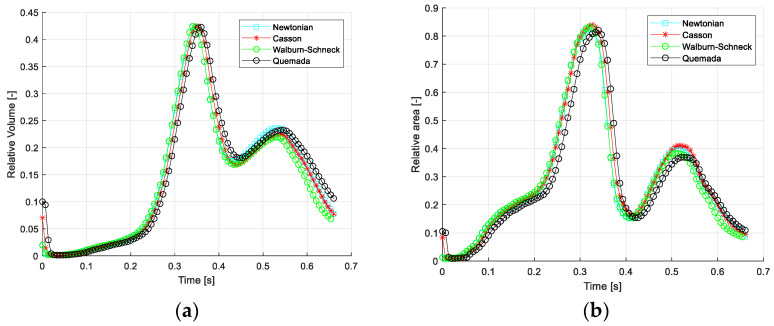
Relative retrograde flow in the thoracic aorta vs. cardiac cycle: (**a**) relative volumetric flow and (**b**) negative WSS area.

**Figure 5 bioengineering-10-01240-f005:**
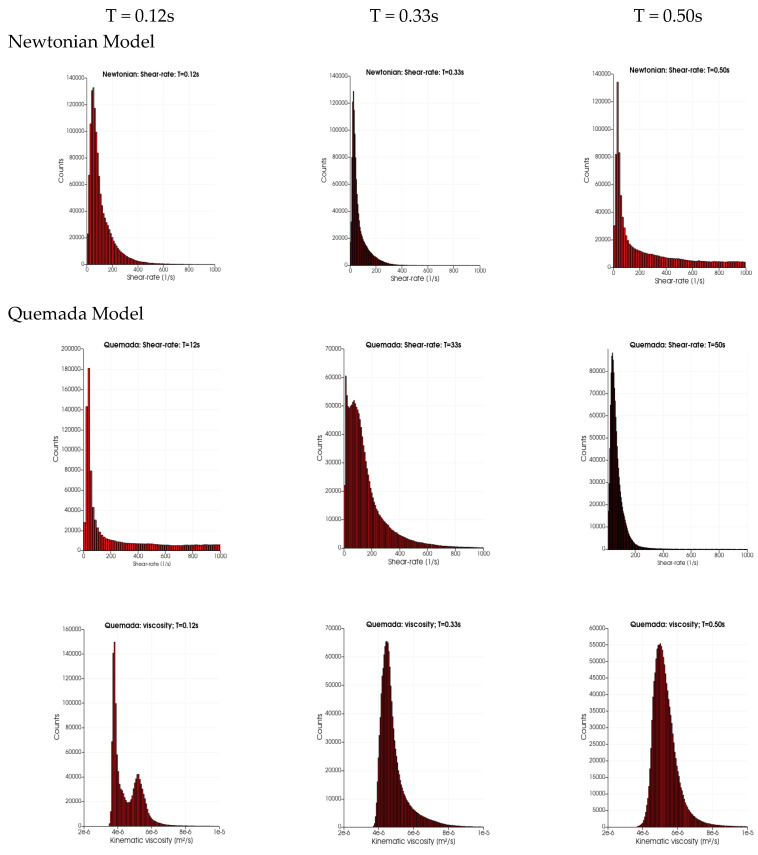
The distribution of shear rate (SR) and kinematic viscosity for the different rheological models at different instants in the cardiac cycle: T = 0.12 s (peak systole), end-systole (T = 0.33 s) and mid-diastole (T = 0.50 s).

**Figure 6 bioengineering-10-01240-f006:**
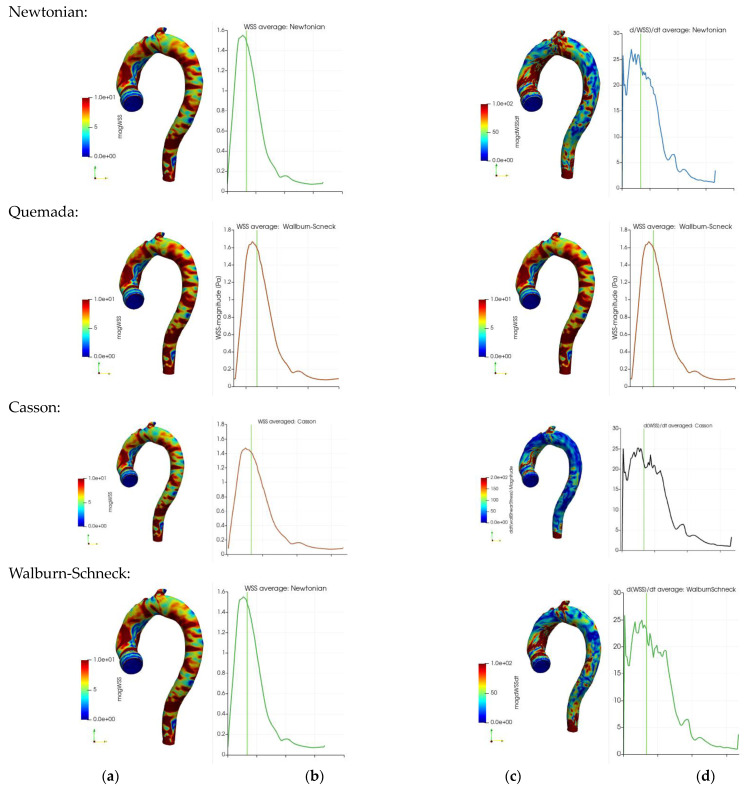
Wall shear stress (Pa) and its temporal derivative (Pa s^−1^) for the four rheological models: instantaneous WSS distribution at near peak systole (frames (**a**,**c**)). Surface average of the time-derivative of WSS vs. the cardiac cycle (frames (**b**,**d**)). The vertical green lines in frames b and d mark the time instant of the WSS in frames b.

**Table 1 bioengineering-10-01240-t001:** Time and space statistical values for *I_L_* and *I_g_*, with the three non-Newtonian models.

Model	Quemada		Casson		Walburn–Schneck	
	*I_L_*	*I_g_*	*I_L_*	*I_g_*	*I_L_*	*I_g_*
Mean	1.536	0.538	1.215	0.215	1.690	0.631
Std	0.288	0.266	0.166	0.158	0.290	0.535
Max	5.755	4.745	6.002	4.991	4.480	14.352
Min	0.817	0.011	0.906	0.017	0.876	0.009

**Table 2 bioengineering-10-01240-t002:** The mean and RMS of spatial and temporal statistics of WSS and its time derivative.

Model	|WSS| (Pa)	$\left|\frac{\partial WSS}{\partial t}\right|$ (Pa/s)
Newtonian	2.122 ± 3.294	36.349 ± 115.49
Quemada	2.044 ± 2.678	32.251 ± 90.47
Casson	2.045 ± 2.994	35.058 ± 112.17
Walburn–Schneck	2.049 ± 3.145	46.360 ± 171.82

**Table 3 bioengineering-10-01240-t003:** The space-time averages of three WSS indicators demonstrate the marginal effects of the rheological models on these parameters.

Model	OSI (-)	TAWSS (Pa)	RRT (s)
Newtonian	0.112 ± 0.100	3.146 ± 4.889	0.0027 ± 0.0049
Quemada	0.105 ± 0.100	3.023 ± 3.975	0.0020 ± 0.0033
Casson	0.113 ± 0.100	3.031 ± 4.443	0.0026 ± 0.0049
Walburn–Schneck	0.112 ± 0.100	3.032 ± 4.667	0.0026 ± 0.0046

**Table 4 bioengineering-10-01240-t004:** The time- and space-averaged retrograde flow volume and the corresponding negative WSS.

Model	Relative Retrograde Flow Volume		Relative WSS < 0 area	
	Average	Peak	Average	Peak
Newtonian	0.10240	0.36405	0.21834	0.82772
Quemada	0.10476	0.36308	0.20845	0.82053
Casson	0.10487	0.36605	0.22388	0.83970
Walburn–Schneck	0.10445	0.36578	0.22001	0.83248

**Table 5 bioengineering-10-01240-t005:** The “age” of particles leaving at the distal exit of the thoracic aorta (entrance to the abdominal aorta, denoted as EXT). The age is given in seconds. As the heart rate was 90 BPM, the corresponding values in terms of cardiac cycle may be obtained by multiplying the figures in the table by a factor of 1.5.

Model	Particle Diameter = 2 mm (s)				Particle Diameter = 2 mm (s)			
	Mean	Std	Max	Min	Mean	STD	Max	Min
Newtonian	1.196	0.269	3.022	0.654	1.150	0.339	3.755	0.680
Quemada	1.211	0.277	2.637	0.660	1.157	0.348	3.929	0.667
Casson	1.209	0.277	2.845	0.663	1.143	0.334	3.933	0.673
Walburn–Schneck	1.197	0.270	3.115	0.654	1.138	0.327	3.873	0.679

**Table 6 bioengineering-10-01240-t006:** The “age” of particles leaving at the distal exit of the brachiocephalic artery (BC).

Model	Particle Diameter = 2 mm (s)				Particle Diameter = 2 mm (s)			
	Mean	Std	Max	Min	Mean	Std	Max	Min
Newtonian	0.916	0.268	2.724	0.205	0.895	0.308	3.696	0.325
Quemada	0.871	0.350	5.265	0.287	0.843	0.277	2.856	0.160
Casson	0.879	0.310	3.993	0.278	0.866	0.265	2.768	0.167
Walburn–Schneck	0.895	0.267	2.909	0.175	0.850	0.307	3.742	0.256

**Table 7 bioengineering-10-01240-t007:** The “age” of particles leaving at the distal exit of the left common carotid artery (LC).

Model	Particle Diameter = 2 mm (s)				Particle Diameter = 2 mm (s)			
	Mean	Std	Max	Min	Mean	Std	Max	Min
Newtonian	0.762	0.290	2.129	0.141	0.634	0.307	3.657	0.194
Quemada	0.706	0.276	2.134	0.147	0.641	0.324	5.292	0.182
Casson	0.751	0.290	2.397	0.137	0.639	0.287	3.562	0.189
Walburn–Schneck	0.739	0.271	2.055	0.147	0.628	0.294	3.373	0.188

**Table 8 bioengineering-10-01240-t008:** The “age” of particles leaving at the distal exit from the left subclavian arteries (LS).

Model	Particle Diameter = 2 mm (s)				Particle Diameter = 2 mm (s)			
	Mean	Std	Max	Min	Mean	Std	Max	Min
Newtonian	1.164	0.368	3.819	0.429	1.072	0.356	3.749	0.644
Quemada	1.149	0.344	3.884	0.240	1.103	0.409	5.970	0.635
Casson	1.132	0.308	3.604	0.472	1.071	0.331	3.771	0.637
Walburn–Schneck	1.134	0.312	3.175	0.612	1.069	0.329	3.905	0.640

## Data Availability

Data are available upon request to the corresponding author.

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
