# Peer review of "Assessment of Rheological Models Applied to Blood Flow in Human Thoracic Aorta"

_bioengineering, 2023, doi:10.3390/bioengineering10111240_

Round 1
Reviewer 1 Report (Previous Reviewer 1)
As I have found that the authors have reasonably responded to my previous comments, I have no further objections for their work to be published.
Author Response
We thank the reviewer, and happy the reviewer finds the response adequate.
Reviewer 2 Report (Previous Reviewer 2)
In this articles the authors studied the importance of non-Newtonian rheological models on blood flow in the human thoracic aorta. The pulsatile flow in the aorta is simulated using the models of Casson, Quemada and Walburn-Schneck in addition to a case of fixed (Newtonian) viscosity.
I recommended a minor revision. The observations are:
1. The abstract must be short and to the point in one paragraph.
2. Blood is Newtonian or non-Newtonian fluid?
3. The introduction section must modify and consider some recent results. I recommended considering "Significance of body acceleration and gold nanoparticles through blood flow in an uneven/composite inclined stenosis artery: A finite difference computation".
4. Give reference to Equations.
5. Compare your results with published work, if possible (optional).
Moderate editing of the English language required
Author Response
We thank the reviewer for the comments and hope the response is found adequate.
- The abstract must be short and to the point in one paragraph.
The abstract has been revised, considering both this response and previous review comment asking for extended abstract.
- Blood is Newtonian or non-Newtonian fluid?
Just as stated in the first line of introduction, blood is a non-Newtonian fluid.
- The introduction section must modify and consider some recent results. I recommended considering "Significance of body acceleration and gold nanoparticles through blood flow in an uneven/composite inclined stenosis artery: A finite difference computation".
The relevance to blood flow is very low and hence we do not find it relevant to this paper.
- Give reference to Equations.
We have gone through all equations, and all references are now included.
- Compare your results with published work, if possible (optional).
- Comments on the Quality of English Language: Moderate editing of the English language required.
We have thoroughly read the manuscript and revised it for language.
Reviewer 3 Report (Previous Reviewer 3)
Authors have performed the changes suggested by this reviewer. In my opinion, the article may be considered for acceptance.
Author Response
We thank the reviewer, and happy the reviewer finds the response adequate.
This manuscript is a resubmission of an earlier submission. The following is a list of the peer review reports and author responses from that submission.
Round 1
Reviewer 1 Report
In this work the authors offer a computational study of the blood flow within the human thoracic aorta. 3 different generalized non-Newtonian models are used in addition to the Newtonian model. The results are compared in terms of 4 different measures involving the flow structure as well the wall shear stress and the distribution of particles dispersed in the flow. Most differences are observed in terms of the maximum residence time for the dispersed particles in the flow.
There are several problems with this work as discussed in the detailed comments below. In addition, as admitted by the authors themselves (starting at line 529 …) “For large shear-rates (depicted in Fig 2a), the differences among the models are considerably smaller than for shear-rates below O(100 530 s-1). Hence, for predominantly high shear-rate flows, assuming that the blood mixture is a Newtonian fluid is a natural choice” thereby making the usefulness of this study applied here to a major artery of little value. Especially concerned I am with the artificial nature of the outlet boundary conditions used in the present work that even more significantly limit the usefulness of their study. My recommendation is therefore to reject this manuscript for publication.
Detailed comments
1. The authors need to justify the outlet boundary conditions used in their work. One of the key impacts of a variable rheology being the redistribution of the flow, it is very restrictive to consider the flow distribution to the various outlets as given in all situations. Much more appropriate are lumped parameter/Windkessel (0D) model to a 1D full arterial network conditions, as for example that employed in [1]. I can see very little usefulness with results obtained with the conditions used in the paper.
2. The parameters of the four different stress models employed in this work are discussed in section 2.1 but not justified. A proper justification is needed (fitting, for example, data) as well as though a stress vs. shear rate plot where the different model predictions are indicated. This is important in order to ascertain how reasonable the comparison among the four different models is.
3. Starting from line 184 the shear rate is not denoted properly (with a dot over the letter gamma).
4. The numerical parameters used are not properly justified. A mesh refinement study in at least one representative case is needed to demonstrate convergence and illustrate the magnitude of the errors expected from the numerical discretization.
References
[1] Apostolidis, A.J., Moyer, A.P., Beris A.N., “Non-Newtonian effects in simulations of coronary arterial blood flow.” J. Non-Newtonian Fluid Mech., 233, 155-165 (2016).
Reviewer 2 Report
The studied subject is interesting and has applications in real-world problems. I recommended a minor revision after the following observations:
1. The abstract must modify and I do not agree to add more explanations in the abstract. It must be to the point.
2. The abstract section must modify and consider some recent results using mathematical concepts with applications.
3. Effects of the fractional order and magnetic field on the blood flow in cylindrical domains.
4. Give references to Eqs. (1) and (2).
Minor editing of English language required
Reviewer 3 Report
In this work authors evaluate the goodness of different existing non-Newtonian rheological equations to model the blood flow in human thoracic aorta. The article is really interesting and results are potentially useful for research community. My comments are written as follows:
1) The abstract merely describes what was done in the research. Authors should include any numerical conclusion to better highlight the outcome of their investigation. Likewise, conclusions (at the end of the article) could be written in a more comprehensive way.
2) A table summarizing the main characteristics of assessed models would be of great help for the reader.
3) Some assumptions should be supported by appropriate literature references. For instance, blood was considered incompressible and the concentration of red blood cells was assumed to be fix. How was the value of hematocrit selected to α = 0.45? Similar with the boundary conditions.
4) Provide complete information regarding the used simulation packet.
5) Authors claim in their article that obtained conclusions are generally valid. However, reported results are limited to a single geometry (and a single heat rate). The article should provide more reasoning regarding how these characteristic may affect the effectiveness of existing rheological models.
6) Some more literature comparison and contrast would be desirable when discussing results.
Other minor remarks are: i) Revise unnecessary symbol in table 2.